# Leptospirosis in the Caribbean Region between 2000 and 2022: A scoping review of morbidity and mortality

Beatris Mario Martin[1]*, Zhonghan Zhang[1], Sebastian Vernal[1], Holly Jian[1], Eric J. Nilles[2,3,4], Luis Furuya-Kanamori[1], Benn Sartorius[1], Colleen L. Lau[1]

1 University of Queensland Centre for Clinical Research (UQCCR), Faculty of Health, Medicine, and Behavioural Sciences, The University of Queensland, Brisbane, Australia, 2 Harvard Humanitarian Initiative, Cambridge, Massachusetts, United States of America, 3 Harvard Medical School, Boston, Massachusetts, United States of America, 4 Brigham and Women's Hospital, Boston, Massachusetts, United States of America

* beatris.martin@uq.edu.au

## Abstract

### Background

Leptospirosis remains an important yet underreported public health concern in the Caribbean. Urbanisation, biodiversity loss and human encroachment into natural habitats have contributed to shifts in its epidemiological patterns. However, accurate assessment of disease burden is hindered by limited diagnostic capacity, surveillance challenges, and scarce research. We aim to describe geographical and temporal distribution of leptospirosis epidemiology in Caribbean Island Countries and Territories (CRICTs) and identify patterns and gaps in knowledge.

### Methodology/principal findings

We conducted a systematic search across PubMed, Web of Science, Embase, Scopus, and the Latin America and Caribbean Health Sciences Literature databases, between 2000–2022, without language restrictions. Eligible publications were routine surveillance-based studies or cross-sectional seroprevalence surveys. We followed the Preferred Reporting Items for Systematic Reviews and Meta-analyses extension for Scoping Reviews (PRISMA-ScR) protocol. Of 110 full-text articles reviewed, 16 met inclusion criteria, documenting leptospirosis in 15 of 27 CRICTs (55.6%). Between 2000–2010, we identified an average of 2.6 studies per year, compared to just 1.2 between 2011–2022. Nine studies (60.0%) reported surveillance data, and six (40.0%) were seroprevalence surveys. Two studies reported hospitalisation rate (12.5%), and five studies, case fatality rate (31.3%). There were more than one publication from Guadeloupe, Jamaica, Puerto Rico, St. Lucia and Trinidad and Tobago.

**Data availability statement:** All data analysed in the manuscript were obtained from publicly available sources, and source details are provided throughout the manuscript and supporting documents. A repository of all extracted data is available in https://github.com/BeatrisMartin/Leptospirosis-Caribbean-Scoping-Review.

**Funding:** 1. ZZ received a UQ scholarship (summer project). BMM received a UQ Research Training Stipend Scholarship. CLL was supported by an NHMRC Investigator Grant (APP1193826). BS was supported by an NHMRC Investigator Grant (GNT2034827). LFK was supported by the University of Queensland's Amplify Initiative. The funders had no role in study design, data collection and analysis, decision to publish, or preparation of the manuscript.

**Competing interests:** The authors have declared that no competing interests exist.

Although most studies acknowledged links between leptospirosis and extreme weather, only three were specifically designed to investigate this association.

## Conclusions/significance

Our findings highlight critical gaps in leptospirosis burden and research across the Caribbean. The scarcity of recent studies investigating epidemiological differences across rural and urban settings, and the impact of environmental changes, contributes to limited characterisation of evolving transmission patterns across the region. Strengthening regional research capacity and surveillance systems is essential to inform targeted public health strategies and reduce the disease's burden locally.

## Author summary

Leptospirosis continues to affect communities across the Caribbean. Growing cities and population, closer contact between people and wildlife, and more frequent heavy rain and flooding can facilitate *Leptospira* survival and transmission. Despite its importance, the most recent regional estimates were published over a decade ago and relied on information that largely predates the year 2000. This limits the effectiveness of public health responses and disease control strategies. To address this gap, we conducted a scoping review of studies published between 2000–2022 to provide an updated overview of leptospirosis epidemiology across Caribbean Islands Countries and Territories (CRICTs). We identified 16 studies reporting data from 15 of 27 CRICTs, revealing substantial gaps in surveillance, geographic coverage, and research on environmental drivers. Our findings underscore the need for strengthened regional research capacity and improved surveillance systems to better understand evolving transmission patterns and inform targeted interventions to reduce the burden of leptospirosis in the Caribbean.

## Introduction

Leptospirosis is a zoonotic disease caused by pathogenic strains of the *Leptospira* bacteria [1], affecting both humans and animals, with significant consequences for public health. Globally, it is estimated to cause 2.9 million years of lives lost annually [2] and, in 2019, the human productivity cost of leptospirosis was estimated at 29.3 billion international USD [3]. Although humans are an incidental host and do not contribute to the transmission cycle, they are highly susceptible to infection. Severe forms of the disease, such as pulmonary haemorrhage, may reach a case fatality rate (CFR) of up to 40% [4]. The most recent global estimates for morbidity and mortality published in 2015 by Costa et al. [5] estimated 1.03 million symptomatic human leptospirosis cases and nearly 60,000 related deaths annually. However, these figures likely underestimate the true burden due to limitations in access to

health care, laboratory diagnostic availability and surveillance systems' capability, particularly in tropical resource-poor countries and rural communities where leptospirosis poses a significant burden [3,5]. Moreover, studies representing the Caribbean Region (CR) included in calculating these estimates were based on studies published before 2004, reporting data predominantly collected before the early 2000s and therefore do not reflect more recent shifts in disease burden and transmission dynamics.

Accurate diagnosis and reporting of leptospirosis remain complex due to various challenges. While multiple diagnostic tests exist, laboratory-confirmed diagnosis relies mainly on serological methods. The gold standard serological assay is the microscopic agglutination test (MAT), which is time-consuming, costly, and often unavailable or difficult to access in certain regions [1]. Diagnostic accuracy is further complicated by the need to include locally circulating serovars in the MAT panel [6,7], and interpretation of results can be complicated by cross-reaction between different serogroups and long-term persistence of antibodies [8]. Clinically, leptospirosis often presents as a non-specific febrile illness, indistinguishable from many other tropical endemic diseases, such as dengue, chikungunya or malaria [9–12]. These factors contribute to frequent misdiagnosis and underreporting, obscuring the true epidemiological picture and hindering effective public health responses.

Historically considered an occupational disease affecting agricultural and animal workers, leptospirosis is increasingly reported in urban and peri-urban environments. Recent evidence indicates a shifting epidemiological pattern, driven by urbanisation, biodiversity loss, and human encroachment into natural habitats [13]. These changes have facilitated the proliferation and spread of rodent populations [14,15]–which are competent reservoirs of *Leptospira*–and increased human exposure to contaminated environments. Moreover, extreme weather events, such as floods, intensified by climate change, further elevate transmission risk, particularly in densely populated areas, such as urban and peri-urban settings [16]. These evolving drivers of transmission may not have been captured in the global estimates published by Costa et al. [5], underscoring the need for updated data that reflect recent sociodemographic and environmental conditions.

Tropical small island developing states, such as the countries and territories in the CR, are particularly vulnerable environments for leptospirosis transmission and outbreaks [5,13]. A study investigating leptospirosis outbreaks between 1970 and 2012 identified that 16.4% of global outbreaks were reported in the CR [17]. During this period, globally, rural settings and occupational environments were less frequently associated with the outbreaks. Despite the recognised burden of leptospirosis in the CR, country-level incidence data remain limited and inconsistent. As leptospirosis epidemiology continues to evolve, it is crucial to reassess its current epidemiological landscape. This scoping review aims to describe the geographical and temporal distributions of studies investigating leptospirosis epidemiology in the CR from 2000 onwards, to identify patterns and gaps in data and knowledge.

## Methods

### Ethics statement

All data included in this systematic review were obtained from publicly available sources, and no formal ethical approval was required.

This scoping review followed the Preferred Reporting Items for Systematic Reviews and Meta-analyses extension for scoping reviews (PRISMA-ScR) protocol [18]. The PRISMA-SCR checklist is available in the S1 Table.

### Geographic scope

We limited our scope to island countries and territories located in the Caribbean Sea and excluded continental countries with a Caribbean coast (i.e., Belize, Colombia, Costa Rica, Guatemala, Honduras, Mexico, Nicaragua, Panama, Suriname, and Venezuela), based on the higher risk for leptospirosis transmission associated with small islands [5,13,17]. We identified 27 CR island countries and territories (CRICTs) that were eligible for inclusion: Anguilla, Antigua and Barbuda, Aruba, Bahamas, Barbados, British Virgin Islands, Cayman Islands, Cuba, Curacao, Dominica, Dominican Republic,

Grenada, Guadeloupe, Haiti, Jamaica, Martinique, Montserrat, Puerto Rico, Saint Barthelemy, Saint Kitts and Nevis, Saint Lucia, Saint Marteen, Sain Martin, Saint Vicent and the Grenadines, Trinidad and Tobago, Turks and Caicos, and U.S. Virgin Islands (**Fig 1**).

## Search strategy

On 17 February 2025, we conducted a systematic search of five databases (PubMed, Web of Science, Embase, Scopus and Latin America and Caribbean Health Literature (LILACS)) for publications reporting on leptospirosis epidemiology based on data collected from 2000 onwards. In each database, the search term applied was a combination of *'Leptospirosis'* AND *'Country/territory'*. During title and abstract screening, no time or language restrictions were imposed. We conducted a backward citation search to identify additional peer-reviewed studies. A full description of search strategies for each database and the number of publications retrieved can be found in the S2 Table.

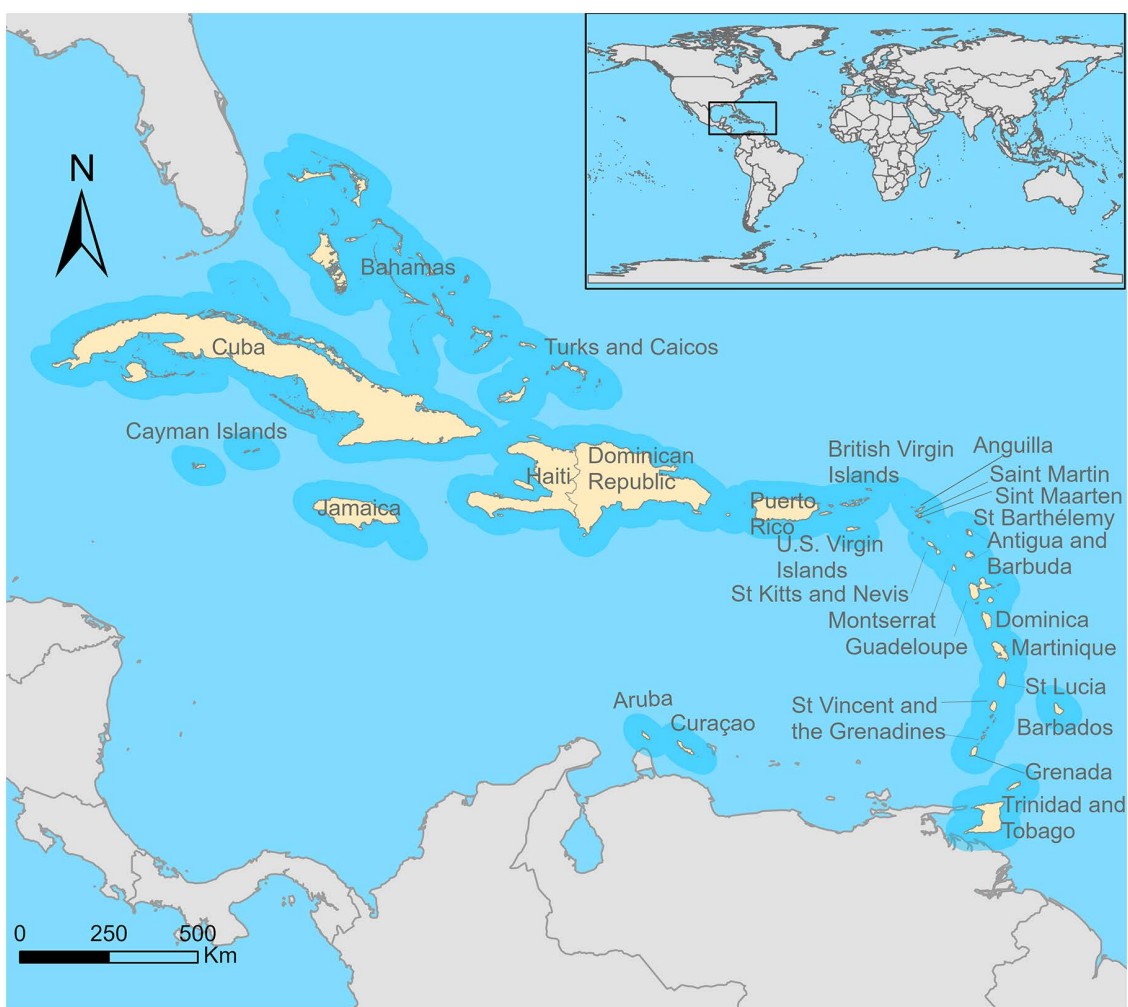

**Fig 1. Caribbean Region island countries and territories.** In light yellow, the 27 Caribbean Region countries and territories included in our review. Other countries are in light grey. Base layer from: https://datacatalog.worldbank.org/search/dataset/0038272/world-bank-official-boundaries.

## Eligibility criteria

**Inclusion criteria.** We included studies reporting on incidence, prevalence, and/or mortality as part of the screening process. A publication was included in this review if the location of the study included at least one of the 27 CRICTs AND reported (i) the total number of identified leptospirosis cases (no restrictions in the diagnostic procedures) in a defined population (national or sub-national level) and period, or (ii) leptospirosis incidence (cases/ 100,000 population) at the national or sub-national level, or (iii) leptospirosis seroprevalence in a defined population and period.

**Exclusion criteria.** We excluded publications focusing on (i) animal leptospirosis, (ii) experimental data (in vitro or in vivo cellular, molecular, biochemical or other studies that did not report on the natural occurrence of leptospirosis in humans), and (iii) studies of laboratory methods.

## Screening and data extraction

First, retrieved publications were uploaded to Rayyan [19], where one reviewer (ZZ) excluded all duplicates. Second, the remaining titles and abstracts were independently screened by two reviewers (ZZ and BMM), and all publications that met the inclusion criteria underwent full-text review. Third, full-text versions in English were assessed for eligibility by the same two reviewers (ZZ and BMM), and Spanish and French publications were assessed by two other reviewers (SV and LFK, and HJ and BMM, respectively). All publications for which no abstract was available were included in the full review screening. Disagreements were resolved by discussion between the two reviewers. If it could not be resolved, a third reviewer was consulted.

Following the inclusion/exclusion process, data were extracted from each of the included publications and recorded in a Microsoft Excel (Microsoft Corporation, Redmond, Washington, USA) [20] spreadsheet by ZZ and cross-checked by BMM. Publications were categorised based on their primary data sources: (i) routine surveillance systems-based studies, which reported cases or population-level incidence obtained from existing surveillance systems (e.g., clinical and laboratory reporting, or systematic notification of suspected cases), and (ii) seroprevalence studies, which reported seroprevalence based on the detection of anti-*Leptospira* antibodies (Ab). Data extracted included: general publication information (authors, title and year of publication), location where the study was conducted, year(s) when data were collected, aims, sampling and study design, characteristics of the study population included (age, gender), number of individuals tested, cases/incidence/prevalence reported, hospitalisation rate, CFR, diagnostic criteria, laboratory test used (MAT, ELISA IgM or IgG detection and polymerase chain reaction (PCR) and serovars identified. For publications that reported both observed and modelled estimates of incidence/prevalence, we extracted the observed data.

## Data analysis

This scoping review focused on providing a comprehensive description of the incidence, prevalence, hospitalisation rate and mortality of leptospirosis in the CR by collating data from multiple study designs and sampling strategies. Publications were aggregated by data source and by location. The aims reported by publications were categorised into five groups: 1) to describe incidence or prevalence of leptospirosis in a country or territory, 2) to identify the proportion of confirmed cases among suspected or symptomatic cases, 3) to identify risk factors, 4) to describe changes in epidemiological pattern over time, and 5) to assess the impact of public health interventions (e.g., antibiotic prophylaxis). For studies reporting more than one aim, each aim was classified according to the above, and all reported aims were included in this analysis.

For studies reporting the number of cases in a defined population over a specified time, we calculated the annual incidence (cases/100,000 population). Population data by CRICT and year were obtained from the World Bank Group [21].

## Results

### Study selection

Our initial search retrieved 1,706 publications, of which 240 were duplicates, and 1,466 publications were screened based on title and abstract. Of these, 110 publications underwent full-text review. We identified 49 publications reporting leptospirosis cases or seroprevalence in at least one of the 27 CRICTs (S3 Table), of which one publication was identified through backward citation search and met all inclusion criteria. Seventeen publications reported data from 2000 onwards; however, one publication was excluded because it was not possible to separate data from before and after the year 2000 [22]. Finally, 16 publications were included in this review (Fig 2). The full list of studies included in this review can be found in the S3 Table.

The 16 publications included in this scoping review documented leptospirosis cases across 15 of the 27 CRICTs (55.6%). Two studies reported cases from multiple locations (12.5%); among these studies, one provided data from eight CRICTs [23], representing 29.6% of the 27 CRICTs and 53.3% of the 15 CRICTs documented in this scoping review. The

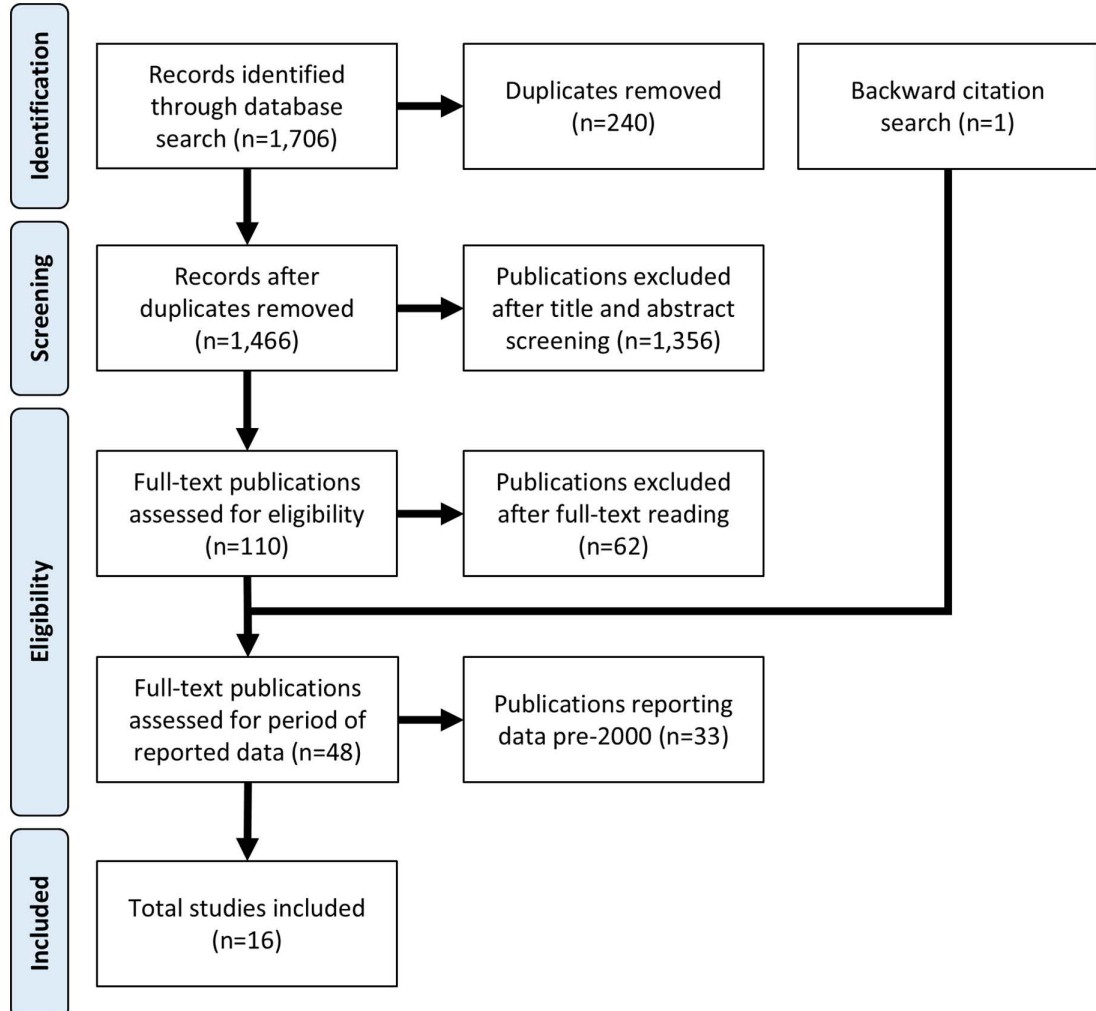

**Fig 2. Flow chart showing the procedure used to search and select studies, and the number of studies included at each stage.**

other multilocation study reported on the epidemiology of leptospirosis in two French overseas territories, Guadeloupe and Martinique [24].

Nine studies reported cases obtained from routine surveillance data (56.3%) as the primary data source, and seven reported seroprevalence studies (43.7%). Routine surveillance-based studies report data from seven CRICTs, representing 25.9% of the 27 CRICTs targeted and 46.6% of the 15 CRICTs documented in this scoping review. Seroprevalence studies were conducted in 12 CRICTs, representing 44.4% of the targeted locations and 80.0% of the included locations.

Nine of the 16 studies reported data from multiple years (56.3%). Regarding the number of studies that reported data from each year, 2007, 2010 and 2011 had the highest frequency, with four studies each, while two years (2018 and 2020) had no reports. During the first half of our study period (2000–2010), an average of 2.6 studies were published per year, whereas in the second half (2011–2022), this number decreased to 1.2 per year (S4 Table). The mean number of routine surveillance-based studies during the first half of our study period was 2.1 per year, compared to 0.75 in the second half. The mean number of seroprevalence studies was 0.55 and 0.41 per year during the first and second halves of our study period, respectively.

## Study characteristics

**Routine surveillance-based studies.** Among the routine surveillance-based studies, the reporting site was hospital in two (22.2%) studies [25,26], and a mix of community and hospital sites in seven (77.8%) studies. All routine surveillance-based studies included cases from urban and rural settings. Case detection among routine surveillance-based studies was passive in all but one study, in which reporting of leptospirosis was complemented by screening of (i) samples from individuals suspected of but negative for dengue and (ii) acute febrile illness fatal cases [27].

Three routine surveillance-based studies mentioned being motivated by the occurrence of a specific extreme weather event (33.3%) [28–30], describing recurrent heavy rainfall and hurricanes causing floods as part of the rationale for conducting the study. Two studies mentioned the strong association between climate (rainfall, wet season, El Niño/La Niña phenomena), without highlighting a specific extreme weather event [26,31]. Six of the surveillance-based studies investigated risk factors associated with incidence (66.7%), five focused on the association between incidence and drivers of transmission (i.e., heavy rainfall, followed by floods) (55.5%) [25,26,29,31,32], and two investigated individual risk factors (22.2%), such as occupational risk, contact with animals, freshwater exposure and household setting (rural) [25,28]. Five routine surveillance-based studies aimed to describe leptospirosis incidence (55.5%) [24,26,29,31,32], four aimed to identify the proportion of suspected cases that were laboratory-confirmed (44.4%) [25,27–29], two aimed to investigate how the incidence changed over time (22.2%) [25,32] and one assessed public health intervention effectiveness (11.1%) [30]. A full description of the characteristics and aims of the routine surveillance-based studies can be found in the S5 Table.

Demographic data were often incomplete, with only three studies reporting age distribution and only two reporting gender distribution. The average incidence per study varied across the CR, with the lowest mean incidence documented in Trinidad and Tobago (2000–2007) and the highest in Guadeloupe (2011) (Table 1). The proportion of laboratory-confirmed cases among suspected cases also varied. Puerto Rico (2011) documented the lowest rate, 8.7%, while St. Lucia (2010–2017) the highest, 40.6%.

**Seroprevalence studies.** Of the studies that documented recruitment site, setting and sampling design, five reported community-based recruitment (71.4%) [33–37], and one reported a mix of community and hospital recruitment (14.3%) [38]; one survey was conducted in a rural setting (14.3%) [36], one in an urban setting (14.3%) [37], and four were conducted in both rural and urban settings (57.1%) [33–35,38]. Three seroprevalence studies used a convenience sampling design [35–37], two a stratified, hierarchical random selection [33,34] and one a simple randomisation [38]. One study did not mention participant recruitment site, setting, sampling design and demographic distribution (14.3%) [23].

Four seroprevalence studies were conducted at the national level (57.1%) [33,34,36,38], and two were conducted at the subnational level (28.6%) [35,37]. All seroprevalence studies aimed to describe the seroprevalence of leptospirosis

**Table 1. Demographics and leptospirosis incidence extracted from routine surveillance-based studies conducted in the Caribbean Region, post-2000.**

| Country/ territory | Study years | Study characteristics | | | Population | | Epidemiology | | |
|---|---|---|---|---|---|---|---|---|---|
| | | Setting[a] | Reporting site[b] | Weather event | Age (Mean) | Males n (%) | Incidence[c] (Mean) | Lab-confirmed rate[d] (%) | Ref |
| Cuba | 2000-2008 | Mixed | Mixed | Hurricane (2007) | NR | NR | 4.2 | NR | [30] |
| Guadeloupe | 2000[e] | Mixed | Hospital | No | 43.3 (18.7) | NR | 4.8[f] | NR | [25] |
| | 2003-2004 | Mixed | Hospital | No | 43.3 (1.2) | NR | 20.0[f] | 33.8 | [26] |
| | 2011 | Mixed | Mixed | No | 40.1 | 393 (66.1) | 30.8 | 21.2 | [24] |
| Jamaica | 2000-2007[e] | Mixed | Mixed | No | NR | NR | 4.5 | NR | [31] |
| Martinique | 2011 | Mixed | Mixed | No | 40.1 | 457 (64.4) | 27.4 | 15.2 | [24] |
| Puerto Rico | 2010 | Mixed | Mixed | No | NR | NR | 4.7 | 8.7 | [27] |
| | 2022 | Mixed | Mixed | Hurricane Fiona (2022) | NR | NR | 8.2 | NR | [28] |
| Saint Lucia | 2010-2017 | Mixed | Mixed | Recurring events[g] | NR | NR | 10.4 | 40.6 | [29] |
| Trinidad and Tobago | 2000-2007[b] | NR | NR | No | NR | NR | 1.6 | NR | [32] |

[a]Setting categories were rural, urban or mixed. [b]Reporting site categories were community, hospital or mixed. [c]Incidence (cases/100,000 people) was calculated based on the number of reported cases by year and the CRICT population according to WorldBank [21]. [d]Criteria used by each study to define laboratory confirmed cases is presented in S7 Table. [e]The total period in which the study was conducted includes years before 2000; however, only results from 2000 onwards included here. [f]Does not represent the totality of cases in Guadeloupe during the study period. [g]Hurricane Tomas (2010), Christmas Eve trough (2012), heavy rainfall associated with floods in 2011 and 2013, tropical storm Mathew (2016) and Hurricane Maria (2017). NR: not reported.

[23,33–38], and three focused on specific groups (42.9%) (i.e., pregnant women [23], veterinary students [35] and sugar-cane fieldworkers [36]). Two seroprevalence studies reported the occurrence of extreme weather events (28.6%) —including hurricanes that occurred in 2017 (Hurricanes Irma and Maria in U.S. Virgin Islands and Hurricane Maria in Puerto Rico) —as part of the rationale for conducting the study [34,37].

Four seroprevalence studies investigated risk factors associated with seroprevalence at the individual level (57.1%) [33–35,37], identified age [33,34], male gender [33], occupation [33–35], animal exposure (cattle [34] and rodents [33,34]), living conditions (household closer to waterbodies and prone to flooding) [37] and household setting (urban) [34] positively associated with seroprevalence. Finally, one seroprevalence study aimed to identify the rate of leptospirosis laboratory-confirmed samples among dengue suspected cases (14.3%) [38]. The full description of the characteristics and aims of the seroprevalence studies can be found in the S6 Table.

Among seroprevalence studies, mean age varied from 21.3 to 42.4 years, and the proportion of males ranged from 20.3% to 91.3%. Excluding the three seroprevalence studies focused on specific groups (i.e., pregnant women, veterinary students and sugarcane fieldworkers), leptospirosis seroprevalence varied considerably across the CR, ranging from 4.1% in the U.S. Virgin Islands (2019) [34] to 27.2% in Puerto Rico (2015) [37] (Table 2).

## Laboratory diagnosis and serovars

Different criteria were used to diagnose leptospirosis cases and measure seroprevalence within and across publications. In the Supplementary information, we provide a full description of the case definitions adopted in each routine surveillance-based study (S7 Table) and the laboratory diagnostic tests used to identify seropositive population in each seroprevalence study (S8 Table).

Among routine surveillance-based studies, all but one provided case definition criteria [30]. Of the eight studies that provided a case definition, ELISA IgM Ab detection was used in all [24–29,31,32], five incorporated MAT results (62.5%)

**Table 2. Demographics and leptospirosis seropositivity extracted from seroprevalence studies conducted in the Caribbean Region, post-2000.**

| Country/ territory | Study characteristics | | | | | Population | | Seropreva-lence | |
|---|---|---|---|---|---|---|---|---|---|
| | Study years | Setting[a] | Recruitment site[b] | Weather event | Sample size | Age (Mean) | Males n (%) | N (%) | Ref |
| **Antigua and Barbuda** | 2009-2011 | NR | NR | No | ≤50[c] | NR | 0[d] | NR (10.5) | [23] |
| **Dominica** | 2009-2011 | NR | NR | No | ≤50[c] | NR | 0[d] | NR (16.0) | [23] |
| **Dominican Republic** | 2021 | Mixed | Community | No | 2091 | NR | 736 (35.2) | 237 (11.3) | [39] |
| **Grenada** | 2009-2011 | NR | NR | No | ≤50[c] | NR | 0[d] | NR (21.6) | [23] |
| **Jamaica** | 2007-2008 | Mixed | Mixed | No | 2419 | 21.3 | NR | 157 (6.5) | [38] |
| | 2009-2011 | NR | NR | No | ≤50[c] | NR | 0[d] | NR (42.6) | [23] |
| **Montserrat** | 2009-2011 | NR | NR | No | ≤50[c] | NR | 0[d] | NR (20.0) | [23] |
| **Puerto Rico** | 2015 | Urban | Community | Hurricane Maria | 202 | 47.8 | 122 (60.4) | 55 (27.2) | [37] |
| **Saint Kitts and Nevis** | 2009-2011 | NR | NR | No | ≤50[c] | NR | 0[d] | NR (9.1) | [23] |
| **Saint Lucia** | 2009-2011 | NR | NR | No | ≤50[c] | NR | 0[d] | NR (19.6) | [23] |
| **Saint Vincent and the Grenadines** | 2009-2011 | NR | NR | No | ≤50[c] | NR | 0[d] | NR (26.0) | [23] |
| **Trinidad and Tobago** | 2006 | Rural | Community | | 704 | 42.4 | 643 (91.3) | 5 (0.7) | [36] |
| | 2010-2011 | Mixed | Community | No | 212 | 23.7 | 43 (20.3) | 12 (5.7) | [35] |
| **U.S. Virgin Islands** | 2019 | Mixed | Community | Hurricane Maria and Irma | 1161 | 42 | 499 (42.9) | 47 (4.1) | [34] |

[a]Setting categories were rural, urban or mixed. [b]Reporting site categories were community, hospital or mixed. [c]An exact sample size was not provided. For each country or territory included, up to 50 pregnant women were enrolled. For this reason, although the seropositivity rate was given, it is not possible to calculate the number of women who tested positive in each country or territory. [d]This study only included pregnant women.

[24–27,31], and three PCR (37.5%) [24,27,35]. Among the seroprevalence studies, four used ELISA IgM [36,38] and IgG Abs [23,35] (57.1%) and four used MAT [33–35,37] (57.1%).

Seven publications (43.8%), including data from five CRICTs (18.5%), reported serogroups based on MAT results, two studies complemented the pathogen identification with molecular tests (PCR and pulse-field gel electrophoresis). Only studies from Puerto Rico [27,37] and Guadeloupe [25,26] reported MAT results in more than one publication. Fig 3 summarises positive serogroups reported by original publications by country. All serovars included in the MAT panel are listed by country/territory in the S9 Table.

## Leptospirosis epidemiology trends over time across Caribbean Region island countries and territories

While leptospirosis epidemiology was investigated in more than one study in five CRICTs (Guadeloupe, Jamaica, Puerto Rico, St. Lucia and Trinidad and Tobago), in seven (Antigua and Barbuda, Grenada, St. Kitts and Nevis, St. Vincent and the Grenadines, and Martinique) it was documented only by one of the multilocation studies (58.3%). Among these CRICTs, six were part of the same multi-location seroprevalence study. Leptospirosis epidemiology data from Guadeloupe, Jamaica and St. Lucia, documented by one of the multilocation studies, could be complemented by other studies. Jamaica and St. Lucia were part of the same seroprevalence multi-location study, reporting seroprevalence among pregnant women, while Guadeloupe, along with Martinique, was part of a routine surveillance-based study, reporting on leptospirosis incidence across two French overseas territories.

**Guadeloupe.** Data from Guadeloupe were complemented by two other publications [25,26], all of which were routine surveillance-based studies. The studies that included data from 2000, 2003, and 2004 had similar case definition criteria, sampling design and methodological approach — reporting on laboratory-confirmed cases among suspected hospitalised

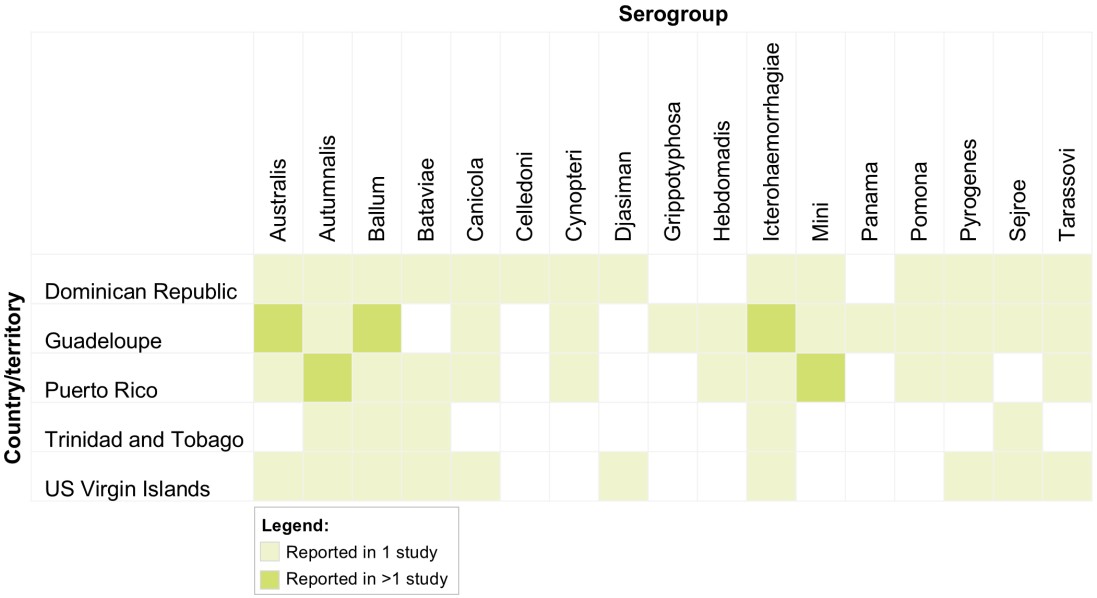

**Fig 3. Serogroups identified reported by country or territory in the Caribbean Region, between 2000 to 2022.**

patients admitted in the largest hospital of Guadeloupe (Hospital of Point-à-Pitre, approximately 80% of all hospital beds in the country). In 2000, 20 confirmed cases were reported (number of suspected cases not reported) [25], followed by 62 (27.9% of all suspected cases) in 2003 and 103 (38.7% of all suspected cases) in 2004 [26]. These results suggested an increasing trend in the number of cases in the early 2000s.The study conducted in 2011 reported 126 confirmed cases (21% of all suspected cases) [24]. This study had a more comprehensive sampling design, including inpatient and outpatient investigations across the entire archipelago. Additionally, it incorporated PCR as a diagnostic tool to enhance case detection and used outpatient data to estimate the 'real' burden (i.e., based on the 126 identified cases, the estimated total number of cases in 2011 was 267 cases; 95%CI 183–351). In our analysis, we only considered the reported cases (126), following the same pattern as other studies.

**Jamaica.** The serosurvey conducted among pregnant women [23] identified Jamaica as having the highest seroprevalence (42.6%) of all CRICTs included in the multi-location seroprevalence study, suggesting a high burden of leptospirosis in the country. Data on leptospirosis epidemiology were further complemented by two studies. A routine surveillance-based study, conducted between 2000 and 2007 [31], showed an increasing trend in reported cases across the years included in the study. In 2005 and 2007, years impacted by La Niña phenomenon, approximately 250 and 320 cases were reported, resulting in 9.3 and 11.7 cases/100,000 population incidence, respectively. Another seroprevalence study, conducted between 2007 and 2008 [38], investigated the proportion of suspected dengue cases that were laboratory-confirmed as leptospirosis. In this study, 6.5% of all samples tested were positive (ELISA IgM Ab) for leptospirosis and 1.9% were positive for both dengue and leptospirosis.

**Puerto Rico.** In Puerto Rico, two studies reported data obtained from routine surveillance systems. One study combined multiple data sources (i-Passive Dengue Surveillance System, ii-Enhanced Fatal Acute Febrile Illness Surveillance System, iii-Health Department reports and iv-two commercial laboratories) to provide a more comprehensive incidence report. By doing so, this study identified 175 cases of leptospirosis in 2010, of which 26 (14.9%) were fatal, whereas routine surveillance had initially only detected 59 cases and two deaths [27]. This study provided an estimated incidence of 4.7 cases per 100,000 population. On September 18, 2022, Hurricane Fiona made landfall in the Southwest

of Puerto Rico. A study investigating the number of reported cases before and after the landfall identified 264 cases in 2022, with the cases reported after the hurricane representing 59.1% of the annual total [28]. Based on this report, in 2022, leptospirosis incidence in Puerto Rico was 8.2/100,000 population. Additionally, a seroprevalence study identified 27.2% seroprevalence based on MAT results (titre >1:50) in a community in San Juan, Puerto Rico [37].

**Saint Lucia.** The seroprevalence of leptospirosis among pregnant women was also documented in St. Lucia, recording 19.6% seropositivity [23]. This was complemented by a routine surveillance study, conducted between 2010 and 2017, varying from 7 cases in 2015 (4.0/100,000 population) to 30 in 2011 (17.5/100,000 population) [29]. No time trend was detected.

**Trinidad and Tobago.** In Trinidad and Tobago, leptospirosis cases were documented by three studies, one based on surveillance data and two in cross-sectional seroprevalence surveys. The surveillance study reported the number of confirmed and probable cases from the national surveillance system from 2000 to 2007 [32]. A peak in leptospirosis cases (36, equivalent to 2.85/100,000 population) was observed in 2005 [32]. The two seroprevalence studies focused on specific demographic groups (university students [35] and sugarcane field-workers [36]), limiting direct comparisons. Furthermore, seroprevalence was determined using distinct laboratory methods; the study on university students [35] used a combination of IgG Ab and MAT, while the study on sugarcane fieldworkers [36] used IgM Ab seropositivity. Worth noting that in the latter, approximately 1% of active sugarcane fieldworkers presented laboratory criteria (positive immunoglobulin M (IgM) Enzyme-linked Immunosorbent Assay (ELISA)) suggestive of acute leptospirosis infection.

### Hospitalisation and case fatality rates

Hospitalisation and CFR were obtained from routine surveillance-based publications. Only two publications reported data regarding hospitalisation [24,40], and five on mortality rate [24,27,29,32,40]. Hospitalisation rate was 79.4% (100 hospitalisations/126 cases) in Guadeloupe, and 64.8% (70/108) in Martinique, both in 2011 [24], and 80.3% (212/264) in Puerto Rico in 2022 [28]. CFR varied between and within CRICTs, ranging from zero (Martinique 2011 [24], St. Lucia 2012–2015 [29], Trinidad and Tobago 2003 and 2008 [32]) to 16.7% (Trinidad and Tobago 2005 [32]). For the CRICTs with reported CFRs, the annual rates are shown in Table 3.

## Discussion

Our findings indicate that more than half of the targeted CRICTs reported leptospirosis cases or seroprevalence in research publications from the year 2000 onwards, highlighting a persistent concern about the disease in the region. However, given the number of countries and territories within the CR and the timeframe of our search, the limited number of retrieved publications suggests constrained research capacity regionally.

In 2017, 18 of the 27 CRICTs experienced floods, hurricanes, and other extreme weather events closely linked to leptospirosis outbreaks [41,42]. Heavy rainfall and flooding increase the risk of leptospirosis by creating conditions that facilitate human exposure to host animals and *Leptospira* bacteria [43,44]. Despite the expected surge in cases following the 2017 extreme weather events in the CR, research documenting their impact remained scarce. Associations between leptospirosis outbreaks and floods and heavy rainfall events have been extensively documented globally [16,43,45]. Although most studies included in this systematic review mentioned the relationship between intense rainfall and floods and leptospirosis, only six [25,26,28,29,31,32] were specifically designed to investigate the impact of rainfall on leptospirosis epidemiology. This limited published literature not only constrains our understanding of important climatic drivers of leptospirosis but also perpetuates the status of leptospirosis as an overlooked disease, where scarce data lead to reduced visibility, ultimately hindering recognition and investment in public health efforts. The sixth Intergovernmental Panel on Climate Change (IPCC) Assessment Report highlighted the CR vulnerability to climate change, intensifying extreme weather events [46]. The absence of a more comprehensive understanding of the current leptospirosis epidemiology limits the region's capacity to anticipate and prepare for future climate-related impacts.

**Table 3. Case fatality rate extracted from routine surveillance-based studies conducted in the Caribbean Region, post-2000.**

| Country/ territory | Study years | Cases | Deaths | CFR (%) |
|---|---|---|---|---|
| **Guadeloupe** | 2011 | 126 | 8 | 6.4 |
| **Martinique** | 2011 | 108 | 0 | 0 |
| **Puerto Rico** | 2010 | 175 | 26 | 14.9 |
| | 2022 | 264 | 25 | 9.5 |
| **Saint Lucia** | 2010 | 17 | 1 | 5.9 |
| | 2011 | 30 | 3 | 10 |
| | 2012 | 11 | 0 | 0 |
| | 2013 | 29 | 0 | 0 |
| | 2014 | 14 | 0 | 0 |
| | 2015 | 7 | 0 | 0 |
| | 2016 | 12 | 1 | 8.3 |
| | 2017 | 25 | 1 | 4.0 |
| **Trinidad and Tobago** | 2001 | 19 | 3 | 15.8 |
| | 2002 | 16 | 1 | 6.3 |
| | 2003 | 23 | 0 | 0 |
| | 2004 | 15 | 1 | 6.7 |
| | 2005 | 12 | 2 | 16.7 |
| | 2006 | 28 | 3 | 10.8 |
| | 2007 | 36 | 1 | 2.8 |
| | 2008 | 17 | 0 | 0 |

We anticipated identifying more surveillance reports of leptospirosis cases in urban settings. In some regions, leptospirosis epidemiology has shifted from a predominantly rural and occupational pattern to a more urban one [47,48] and sporadic recreational exposure [49–52]. Rapid urbanisation, particularly in resource-limited settings, often leads to unplanned and informal settlements with inadequate sanitation and poor waste management, all of which contribute to rodent proliferation and create conditions favourable for human exposure to leptospirosis [14]. The household setting was investigated in two studies [25,34], with opposite results. One study conducted in Guadeloupe during the early 2000s identified a higher risk of confirmed cases among the population living in rural settings [25]. The other, conducted in U.S. Virgin Islands in 2017, identified a higher risk of seropositivity among the urban population [34]. The studies retrieved in this systematic review were evenly distributed across urban and rural settings, and most of the studies reporting data from both settings did not disaggregate data by rural/urban setting. This lack of granularity limits our ability to assess whether urban transmission is indeed increasing in the Caribbean context. Although the number of studies investigating risk factors associated with leptospirosis transmission was limited, both surveillance-based and seroprevalence studies indicated a higher risk associated with occupation and animal exposure [26,28,33–35], which might suggest that work environment context were relevant to leptospirosis transmission in the CR.

Two studies included in this review aimed to report a more 'realistic' incidence, one using a modelling approach [24] and the other by extending leptospirosis screening to other acute febrile illnesses [27]. The raw reported incidence represented between 35.8% to 47.2% of the estimated incidence, indicating an underreporting of leptospirosis cases. Similarly, Nilles et al. used a serocatalytic model to estimate the number of annual leptospirosis infections based on population-level seroprevalence, identifying annual cases nearly 145-fold higher than the average reported cases in the decade that preceded the publication [33]. This potential difference between reported and estimated cases should be considered when analysing data on hospitalisation rates. In our review, although scarce, the hospitalisation rate was high (64.8–80.3%) and

quite homogeneous among the three reporting CRICTs. However, as the number of reported cases most likely excluded mild cases that did not access medical care, and remained undiagnosed and not reported, the case-hospitalisation rate may be overestimated. Additionally, hospitalisation rate can be impacted by several contextual aspects (e.g., access to care, availability of outpatient services and study design), further hindering comparisons across countries and territories, especially when the data are scarce. CFR is frequently used to draw comparisons across countries, as death is a clear, unambiguous event, and vital statistics systems are often more robust than other surveillance systems. Nevertheless, we observed a great variation within the same country and across countries, suggesting that several factors could be impacting leptospirosis outcomes, not only disease severity.

Better understanding circulating serogroups in the region and across countries can help identifying relevant reservoirs and transmission pathways, and may have public health implications (e.g., potential association with disease severity, definition of a regional MAT panel and targeted serovars in vaccines) [53–55]. Unfortunately, serogroup data were available for fewer than half of the included publications and were restricted to a small number of CRICTs [25–27,33–35,37], with repeated reporting occurring only in Puerto Rico and Guadeloupe. Considerable heterogeneity in the composition of the MAT panel further limited comparability, as differences in serovar selection for the MAT panel influence the likelihood of detecting specific serogroups and may underestimate locally circulating strains. Moreover, reliance on ELISA-based screening in routine surveillance means that serogroup information was typically derived from a subset of cases who underwent confirmatory testing, potentially biasing results towards more severe presentations.

Our review has several limitations. To ensure consistency in data quality and case definition, we excluded outbreak reports and grey literature, which could have introduced variability and potential duplication of cases reported. We therefore limited our review to surveillance-based reports and seroprevalence studies. This approach allowed us to combine complementary data sources: routine surveillance-based studies offer insights into temporal trends and case detection, while seroprevalence studies provide a broader picture of infection burden, including asymptomatic cases. Although a high laboratory-confirmed rate could be expected among the routine surveillance-based studies (i.e., among the ill population), the overall low rate identified in our study highlights the diagnostic challenges. Case identification often relies on nonspecific symptoms and signs that overlap with other endemic diseases in tropical regions (e.g., dengue, malaria) [9,10,12], leading to misdiagnosis and underdiagnosis. Moreover, laboratory confirmation is constrained by limited access to high-quality diagnostics, variation in test sensitivity, and delays in sample collection, all of which can reduce positivity among true cases [6]. In contrast, seroprevalence studies detect antibodies that may persist for up to eight years after infection [8], capturing both recent and previous exposures, partially explaining the elevated seropositivity rates observed among community surveys of generally healthy populations. These diagnostic and temporal dynamics underscore the need for caution when comparing positivity rates across study types. Additionally, included publications varied substantially, reflecting differences in sampling design, study populations, diagnostic methods, and reporting practices, which further limits the comparability of findings and their generalisability. Furthermore, the results reported in our manuscript overrepresent the first half of the study period. Publications up to 2010 were based mainly on routine surveillance data, while reports after 2010 predominantly came from seroprevalence studies. Additionally, even among the post-2010 data, most reports were limited to the pre-2015 period, and only three publications contained data spanning from 2015 to 2023 [28,33,34]. The decreasing trend in leptospirosis reports across the CR aligns with historical shifts in disease prioritisation. Earlier publications frequently positioned leptospirosis as more prominent than other infectious diseases, such as dengue. However, recent publications suggest a reversal in this hierarchy. Recent events, such as the Zika virus outbreak, progressively larger dengue epidemics, and the COVID-19 pandemic, may have contributed to the irregular, less robust, and poorly funded support of other endemic disease surveillance, such as leptospirosis. Despite these limitations, this review contributes valuable insights—particularly through the regional characterisation of circulating serogroups. Although only a few studies reported detailed MAT panel results, serovar characterisation enabled cross-location comparisons, which

can be used to support the development of locally tailored MAT panels and inform vaccine strategies. However, the limited number and geographic scope of these studies constrain the robustness of our conclusions.

While there is general recognition of the burden of leptospirosis in the Caribbean, the lack of accessible and reliable information hinders efforts to build local capacity for timely diagnosis, treatment, surveillance, and monitoring. This absence of data is itself evidence of how little is known about leptospirosis in the region and underscores the challenges of planning prevention strategies within an evolving epidemiological landscape. Strengthening leptospirosis surveillance and research can enhance the identification of epidemiological patterns, improve public health preparedness and reduce the burden of leptospirosis in vulnerable populations already affected by multiple health crises.

## Supporting information

**S1 Table. Preferred Reporting Item for Systematic reviews and Meta-analyses extension for Scoping Reviews (PRISMA-ScR) Checklist.**
(DOCX)

**S2 Table. Full description of the search strategy used in each database and the number of publications retrieved.**
(DOCX)

**S3 Table. The complete list of the 48 publications (no time restriction) reporting leptospirosis cases or seroprevalence in at least one of the 27 CRICTs and the period in which the study was conducted.**
(DOCX)

**S4 Table. The complete list of the 16 publications reporting leptospirosis cases or seroprevalence in at least one of the 27 CRICTs after the year 2000 included in the scoping review.**
(DOCX)

**S5 Table. Summary of characteristics of publications based on routine surveillance-based studies.**
(DOCX)

**S6 Table. Summary of characteristics of publications based on seroprevalence studies.**
(DOCX)

**S7 Table. Case definition presented and laboratory tests used to confirm cases presented in routine surveillance-based studies.**
(DOCX)

**S8 Table. Laboratory test used to identify seropositive individuals in seroprevalence studies.**
(DOCX)

**S9 Table. Serovars included in the MAT by country/territory and publication.**
(DOCX)

## Acknowledgments

This work was supported by the Operational Research and Decision Support for Infectious Diseases (ODeSI) program, which is funded by The University of Queensland's Health Research Accelerator (HERA) initiative (2021–2028).

## Author contributions

**Conceptualization:** Beatris Mario Martin, Luis Furuya-Kanamori, Benn Sartorius, Collen L. Lau.

**Data curation:** Beatris Mario Martin.

**Formal analysis:** Beatris Mario Martin.

**Funding acquisition:** Luis Furuya-Kanamori, Benn Sartorius, Collen L. Lau.

**Investigation:** Beatris Mario Martin, Zhonghan Zhang, Sebastian Vernal, Holly Jian.

**Methodology:** Beatris Mario Martin, Luis Furuya-Kanamori, Benn Sartorius, Collen L. Lau.

**Project administration:** Beatris Mario Martin, Luis Furuya-Kanamori, Collen L. Lau.

**Resources:** Luis Furuya-Kanamori, Benn Sartorius, Collen L. Lau.

**Supervision:** Luis Furuya-Kanamori, Benn Sartorius, Collen L. Lau.

**Validation:** Luis Furuya-Kanamori, Benn Sartorius, Collen L. Lau.

**Visualization:** Beatris Mario Martin.

**Writing – original draft:** Beatris Mario Martin.

**Writing – review & editing:** Beatris Mario Martin, Zhonghan Zhang, Sebastian Vernal, Holly Jian, Eric J. Nilles, Luis Furuya-Kanamori, Benn Sartorius, Collen L. Lau.

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
