## [Decision Letter · Decision Letter 0]

26 Nov 2025

Dear Dr. Beatris Mario Martin,

Thank you for submitting your manuscript to PLOS Neglected Tropical Diseases.

After careful consideration, we feel that your study provides valuable data, described with a clear methodology and conclusions. Overall the reviewers' comments are very positive, but some issues need to be addressed before we can accept this manuscript for publication, justifying a minor revision decision. The reviewers have provided detailed comments. Therefore, we invite you to submit a revised version of the manuscript that addresses the points raised during the review process.

Please submit your revised manuscript within 30 days: December 26th. If you will need more time than this to complete your revisions, please reply to this message or contact the journal office at plosntds@plos.org. Response to Reviewers
Revised Manuscript with Track Changes
Manuscript

Shaden Kamhawi

co-Editor-in-Chief

Paul Brindley

co-Editor-in-Chief

**Journal Requirements:**

**Reviewers' comments:**

**Key Review Criteria Required for Acceptance?**

**Methods**

-Are the objectives of the study clearly articulated with a clear testable hypothesis stated?

-Is the study design appropriate to address the stated objectives?

-Is the population clearly described and appropriate for the hypothesis being tested?

-Is the sample size sufficient to ensure adequate power to address the hypothesis being tested?

-Were correct statistical analysis used to support conclusions?

-Are there concerns about ethical or regulatory requirements being met?

Reviewer #1: (No Response)

Reviewer #2: -The objective is clearly articulated with the study's purpose, but does not have and testable hypothesis due to the study design (Scoping review).

-This design is appropriate for their objective: "To describe geographical and temporal distribution of leptospirosis epidemiology in Caribbean Island 23 Countries and Territories (CRICTs) and identify patterns and gaps in knowledge"

-Definition of study population and statistical analysis does not apply to the study (Scoping review)

-Ethical considerations are appropriate for the study type (Scoping review)

Reviewer #3: The article has the aim to describe geographical and temporal distribution of leptospirosis epidemiology in Caribbean Island Countries and Territories (CRICTs) and identify patterns and gaps in knowledge.

The paper conducted a scoping review of studies published between 2000-2022 to provide an updated overview of leptospirosis epidemiology across CRICTs.

The objectives, methodology and data analysis are suitable for the study’s aim.

However, in the section of data analysis, Line 169-175: the authors described that the aims reported by publications were categorised into 6 groups but the detail are provided only 5 groups.

Reviewer #4: The objectives are clearly described, and the study uses the proper methodology by following the PRISM-ScR checklist. In the Search Strategy (Line 129-130), this sentence "In

130 each database, the search term applied was a combination of ‘Leptospirosis’ AND ‘Country/territory’." is not consistent with the Supporting Table 2. It should be "Caribbean", shouldn't it?

**Results**

-Does the analysis presented match the analysis plan?

-Are the results clearly and completely presented?

-Are the figures (Tables, Images) of sufficient quality for clarity?

Reviewer #1: (No Response)

Reviewer #2: -The analysis matches the analysis plan, a description of the evidence.

-Results are clearly and completely presented. However, the authors include result interpretation and make judgments within the results section.

-The figures are (Tables, Images) of sufficient quality for clarity

Reviewer #3: The results are well presented. Tables and figures are appropriate and clear presentation. Titles, legends, columns and rows labels are mostly correct and clear.

Some minor comments on result sections are as follows:

Line 222: Please check the percentage representing 3 routine surveillance-based studies mentioned being motivated by the occurrence of a specific extreme weather event.

Line 254-256: Please check the reference for this data.

Line 264: Please check the correction of data in the main text and the S6 table mentioned about 5 seroprevalence studies investigated risk factors.

Line 273-275: In the sentence “…leptospirosis seroprevalence varied considerably across the CR, ranging from 4.1% in the U.S.Virgin Islands (2019) (28) to 26.9% in Trinidad and Tobago (2010–2011) (32) (Table 2).”

Please check the reference for this data and the range of % seroprevalence. The studies described in S6 table and table 2 are the same studies?

Line 288: Please remove an extra %.

Kind regards,

Figure 3 legend: Please add the figure legend for white box and the year of the publication should be 2000-2023 or 2000-2022.

Line 353-356: Please add the references for the data of Saint Lucia.

S5 table: the column of “Impact of PH intervention”, what is PH stand for?

S5 and S6 table: It will be useful if the reference of each publication can be added.

Reviewer #4: The analysis matches the plan. Most of the results were clearly and completely presented. A few comments in Table 1. There is a missing unit of incidence in Table 1, which is to ensure that the values were adjusted by year and population in the study area. It is not clear to me what you mean by "Not population representative" in the footnote of Table 1. I suggest describing more either in the text or in the footnote. Lastly, although the "Lab confirmed rate" is useful information, the description of tests (used to confirm cases) is also important for demonstrating the case report accuracy in each study. I suggest adding the methodology used for case confirmation in Table 1 and/or in the text.

**Conclusions**

-Are the conclusions supported by the data presented?

-Are the limitations of analysis clearly described?

-Do the authors discuss how these data can be helpful to advance our understanding of the topic under study?

-Is public health relevance addressed?

Reviewer #1: (No Response)

Reviewer #2: -There are some analyses made in the results section after the description of that evidence that are not fully supported by the findings (for example, 319-328 (trend increase in incidence when the 2011 numbers were lower than the 2003 and 2004) and 339-349 (they suggest misdiagnosis without considering coinfection)).

-The authors never make references to the limitations of the studies they are reporting on, and present the results of these studies as generalizable facts.

-The authors outline the importance of their findings, gaps in knowledge around leptospirosis in the insular Caribbean Region.

-Public health relevance is addressed

Reviewer #3: In the discussion, the results are discussed in several aspects including extreme weather associated with leptospirosis epidemiology, leptospirosis cases in rural and urban settings and the number of leptospirosis cases (reported and estimated) in CRICTs. However, some results from data analyses are not discussed such as the risk factors from surveillance-based studies and seroprevalence studies and circulating serogroups presented in CRICTs.

The limitations of the study are clearly described. The conclusions are supported by results and the public health relevance is addressed.

Reviewer #4: Conclusions supported by the data. The limitations are clearly discussed. And the public health relevance is addressed.

**Editorial and Data Presentation Modifications?**

Reviewer #1: (No Response)

Reviewer #2: Given that some of the studies used specific populations (pregnant women, hospitalized patients with clinical suspicion of leptospirosis) and sample sizes (representative?), and that some of the tests used do not constitute diagnostic confirmation, the authors should be careful how they phrase their analysis to make their results represent the whole incidence or hospitalization rate for a CRICT.

There seems to be a few discussion elements in the results section. For example, in the section providing data on the findings in each CRICT, the authors follow up the report of what was found with an interpretation of those results. Such interpretations are usually reserved for the discussion section.

Specific comments:

95-97: The author's statement about 16.4% of global outbreaks taking place in the CR is not supported by the study used as a reference.

170-174: The author mentions 6 groups but only describes 5

215: I think a clear definition of “routine surveillance” might help the reader interpret the authors' findings.

319-328: Can you really talk about a trend increase in incidence when the 2011 numbers were lower than the 2003 and 2004 numbers?

339-349: Can you really reach that conclusion, considering the possibility of coinfections or the evidence that Dengue virus-infected patients can generate false positives in Leptospira spp. MAT?

347-349: You should probably include the month of Hurricane Fiona to make that information more impactful to the reader

366: Please specify the criteria you are alluding to?

424-425: Can study designs have impacted hospitalization rates?

Reviewer #3: (No Response)

Reviewer #4: (No Response)

**Summary and General Comments**

Reviewer #1: (No Response)

Reviewer #2: This is a relevant study that puts into evidence that leptospirosis is not only a neglected but also forgotten tropical disease. Its methodological structure is solid, the paper is well written, and the authors' findings are supported by what was reported. The authors need to be more critical of the evidence they are reporting on, especially when generalize one study's findings as evidence at a national level.

Reviewer #3: The study provides valuable update data of leptospirosis epidemiology in CRICTs although the relevance studies in this region are limited. The results reveal knowledge gaps, offering opportunities to strengthen research and surveillance systems in this region.

Reviewer #4: (No Response)

PLOS authors have the option to publish the peer review history of their article (what does this mean? ). If published, this will include your full peer review and any attached files.

**Do you want your identity to be public for this peer review?** For information about this choice, including consent withdrawal, please see our Privacy Policy .

Reviewer #1: **Yes: ** HERRMANN STORCK Cécile

Reviewer #2: No

Reviewer #3: No

Reviewer #4: No

**Figure resubmission:**

**Reproducibility:** To enhance the reproducibility of your results, we recommend that authors of applicable studies deposit laboratory protocols in protocols.io, where a protocol can be assigned its own identifier (DOI) such that it can be cited independently in the future. Additionally, PLOS ONE offers an option to publish peer-reviewed clinical study protocols. Read more information on sharing protocols at https://plos.org/protocols?utm_medium=editorial-email&utm_source=authorletters&utm_campaign=protocols

---

## [Editor Report · Decision Letter 1]

15 Dec 2025

Thank you for revising the manuscript according to all the reviewers' suggestions.

I believe the manuscript does not need to be checked again by the reviewers and is now suitable for publication.

However, before I can recommend acceptance, I have a few more (minor) comments:

There are a few typos to be corrected:

L248 WorldBank(21). Please add a space before parenthesis

L340 the early 200s. Correct for 2000s

L452 identifying an annual cases... Please correct

L453 preceded the publication(33). Please add a space before parenthesis

L445-446 “which might suggest that living in rural areas and work environment context were relevant to leptospirosis transmission in the CR.” 

The previous paragraph explains the limitations regarding comparing urban/rural settings in the various studies, with sometimes contradictory results. So I would refrain mentioning “living in rural areas” as relevant to leptospirosis transmission when it does not seem you can make any suggestion in either direction.

Shaden Kamhawi

co-Editor-in-Chief

Paul Brindley

co-Editor-in-Chief

**Journal Requirements:**

**Reviewers' comments:**
**Figure resubmission:**

**Reproducibility:** To enhance the reproducibility of your results, we recommend that authors of applicable studies deposit laboratory protocols in protocols.io, where a protocol can be assigned its own identifier (DOI) such that it can be cited independently in the future. Additionally, PLOS ONE offers an option to publish peer-reviewed clinical study protocols. Read more information on sharing protocols at https://plos.org/protocols?utm_medium=editorial-email&utm_source=authorletters&utm_campaign=protocols

---

## [Editor Report · Decision Letter 2]

20 Dec 2025

Dear Dr Martin,

We are pleased to inform you that your manuscript 'Leptospirosis in the Caribbean Region between 2000 and 2022 - a scoping review of morbidity and mortality' has been provisionally accepted for publication in PLOS Neglected Tropical Diseases.

Best regards,

Vanina Guernier-Cambert, Ph.D.

Guest Editor

Elsio Wunder Jr

Section Editor

Shaden Kamhawi

co-Editor-in-Chief

Paul Brindley

co-Editor-in-Chief

---

## [Editor Report · Acceptance letter]

Dear Dr Mario Martin,

We are delighted to inform you that your manuscript, "Leptospirosis in the Caribbean Region between 2000 and 2022 - a scoping review of morbidity and mortality," has been formally accepted for publication in PLOS Neglected Tropical Diseases.

Best regards,

Shaden Kamhawi

co-Editor-in-Chief

Paul Brindley

co-Editor-in-Chief
